# Stoichiometric and Nonstoichiometric Surface Structures of Pyrochlore $Y_2Zr_2O_7$ and Their Relative Stabilities: A First-Principles Investigation

**Huajian Wu [1], Jianning Zhang [1], Yiren Wang [1], Jiacheng Shang [2] and Yong Jiang [1,3,*]**

[1] Key Laboratory for Nonferrous Materials (MOE), School of Materials Science and Engineering, Central South University, Changsha 410083, China; whj_99@163.com (H.W.); nxgyzhangjianning@163.com (J.Z.); yiren.wang@csu.edu.cn (Y.W.)

[2] Science and Technology on Reactor Fuel and Materials Laboratory, Chengdu 610041, China; niansanlang@163.com

[3] State Key Laboratory of Powder Metallurgy, Central South University, Changsha 410083, China

* Correspondence: yjiang@csu.edu.cn

**Abstract:** First-principle total energy calculations were performed to investigate the atomic structures and relative stabilities of two low miller-index surfaces of pyrochlore $Y_2Zr_2O_7$. The stoichiometric $Y_2Zr_2O_7$ (110) and (100) surfaces were predicted, with lowest formation energies of 1.20 and 1.47 J/m$^2$, respectively. Based on a thermodynamic defect model, non-stoichiometric $Y_2Zr_2O_7$ surface energies were further evaluated as a function of environmental oxygen partial pressure ($p_{O2}$) and temperature ($T$). With all of the results, we were able to construct the surface phase diagrams for $T$ = 300 and 1400 K. The strong correlation between the structural stabilities and the surface stoichiometry was revealed as varying $T$ and $p_{O2}$. At a given $T$, the most stable termination of the (110) surfaces would change from a (Y,Zr)−rich (ns−2Y2Zr6O) to O−rich ones (ns−4O_2 and ns−4O_1) as increasing $p_{O2}$, while that of the (100) surfaces would change from the stoichiometric (stoi−1Y1Zr_1) to the O−rich one (ns−5O). The critical $p_{O2}$ value for termination transition moves to its higher end as increasing $T$.

**Keywords:** $Y_2Zr_2O_7$; stoichiometry; surface energy; surface stability; first principles

## 1. Introduction

Both $ZrO_2$ and $Y_2O_3$ have been among the most widely used functional and structural oxides for many decades. It is known that the binary $ZrO_2$-$Y_2O_3$ system has only one ordered phase of δ-$Y_4Zr_3O_{12}$ [1]. However, Schaedler et al. [2] did later suggest that another crystalline phase of $Y_2Zr_2O_7$ can form at pyrolysis temperatures as low as 500 °C, with, more than likely, a defect fluorite structure than a pyrochlore structure. In nano-structured ferritic alloys (NFAs), both metastable pyrochlore $Y_2Zr_2O_7$ and stable δ-$Y_4Zr_3O_{12}$ can be produced as nano-size dispersoids [3]. NFAs have been regarded as a promising candidate for critical structural uses in advanced nuclear reactors for their exceptional combination of high strength and high resistances to creep and radiation damage under exposure to both high temperatures and high irradiation doses [4]. The extraordinary properties of NFAs can largely be attributed to the precipitation of various complex Y-X-O nano-oxides, including Y-Ti-O [4], Y-Al-O [5,6], Y-Zr-O, and Y-Si-O [7], which show excellent coarsening resistance during high temperature annealing for tens of thousands of hours. With extremely high thermal stabilities, they can not only effectively pin dislocations and grain boundaries but can also act as efficient sinks for various irradiation-induced defects and gas bubbles during the long-term high temperature service [8–11].

It is generally accepted that the exact compositions, structures, and number densities of Y-X-O nano-oxides are highly sensitive to alloy compositions. Several experiments have suggested Zr alloying can induce finer nano-oxides in NFAs, such as, for example, Ti + Zr

co-alloying refined Y-Ti-O nano-oxides to less than 10 nm on average [12–14]. Y + Zr co-alloying in FeCrAl-based NFAs can greatly reduce the number of coarse Y-Al-O nano-oxides in favor of finer Y-Zr-O nano-oxides [5,15–17]. One of our previous first principle studies on nano-clusters (NCs) nucleation also suggested Y-Zr-O NCs nucleate in preference to Y-Ti-O and Y-Al-O NCs in ferritic matrix [18]. Clearly, Y-Zr-O nano-oxides show great promises among all of these Y-X-O nano-oxides. In the recent literature, much research has focused on the atomistic characterization of Y-X-O nano-oxide interfaces in NFAs, suggesting that they all have good coherency (being coherent and semi-coherent) with the matrix [3,19,20]. However, to the best of our knowledge, there is still no fundamental understanding of the surface structures and properties of Y-Zr-O nano-oxides, and this constitutes the critical knowledge base for our future investigations on Y-Zr-O nano-oxide interfaces in NFAs.

In this work, a systematic first principle investigation was carried out to assess the atomic structures and related energetics of pyrochlore $Y_2Zr_2O_7$ (110) and (100) surfaces. The surface phase diagrams of pyrochlore $Y_2Zr_2O_7$ were obtained, and the relative structural stabilities were evaluated over various different surface terminations as a function of $p_{O2}$ and $T$. These results can be used to predict the most energy-favored equilibrium (110) and (100) surface structures during the growth of $Y_2Zr_2O_7$ under a given thermodynamic condition, which provides a critical knowledge base for our future investigation on $Y_2Zr_2O_7$ interfaces in NFAs, but they can also be used to improve our understanding of the reactivity and performance of $Y_2Zr_2O_7$ surfaces in various high temperature catalytic applications, such as SOFC.

## 2. Computational Methods

All of the total energy and structural relaxation calculations were performed using the density functional theory (DFT) code–VASP (Vienna ab-initio simulation package) [21], with plane-wave basis sets and periodic boundary conditions. Ion-electron interactions were approximated using the projector-augmented-wave method (PAW) within the frozen-core approximation [22]. Plane-wave basis sets were generated with the valence configurations of $Y\text{-}4s^24p^64d^15s^2$, $Zr\text{-}4s^24p^65s^24d^2$, and $O\text{-}2s^22p^4$. Convergence tests were performed on the kinetic energy cutoffs of plane-waves and the K-mesh sizes for Brillouin-zone integrations. A high energy cutoff of 550 eV and a $4 \times 4 \times 4$ Monkhorst– Pack K-mesh sufficiently converged the total energy within 1 meV/atom. The electronic minimization algorithm was based on the residual minimization method in the direct inversion of iterative subspace (RMM-DIIS). The exchange-correlation (XC)-functionals were tested using local-density approximation (LDA) and the generalized gradient approximation (GGA) with Perdew-Wang-91 (PW91) [23] and the Perdew–Burke–Ernzerhof (PBE) functionals [24]. The optimal choice of XC-functionals was determined by best reproducing the experimental bulk structures and the elastic modulus in the literature. All of the surface supercell calculations adopted a sufficient vacuum thickness of at least 15 Å and a $4 \times 4 \times 1$ Monkhorst–Pack K-mesh for Brillouin-zone integrations. The ground-state atomic surface structures were optimized by minimizing the Hellmann–Feynman forces until the total force was converged to 0.02 eV/Å on each ion.

## 3. Result and Discussions

### 3.1. Bulk Properties

Pyrochlore $Y_2Zr_2O_7$ has a high symmetry (space group $Fd\bar{3}m$, $O_h^7$). Inside the conventional cell, atoms Y, Zr, O, and O' occupy the Wyckoff positions of 16d (1/2, 1/2, 1/2), 16c (0, 0, 0), 48f ($\delta$, 1/8, 1/8), and 8b (3/8, 3/8, 3/8), respectively. In particular, each O-site atom resides inside a (Y, Zr)-tetrahedron formed by two Y and two Zr atoms, and each O'-site atom inside a Y-tetrahedron solely formed by Y atoms. Due to the influence of the neighboring unoccupied Zr-tetrahedrons (solely formed by Zr atoms), each O-site displaces from the (0, 1/8, 1/8) by $\delta$ that defines the only internal free parameter of the conventional unit cell.

Table 1 summarizes our bulk phase calculation results in comparison with other theoretical and experimental data that is available in the literature. Considering that the GGA (or LDA) type DFT calculations tend to overestimate (or underestimate) experimental lattice constants by <1% and underestimate (or overestimate) experimental bulk modulus by <10%, respectively, all our calculation results agree favorably with available theoretical and experimental values. Clearly, the PAW-PW91 yielded the best predictions on all of the lattice parameters and the bulk modulus. The predicted lattice constant differs only slightly from other DFT results [25–27] by 0.2% (PAW-LDA), −0.2% (PAW-PW91), and 0.4% (PAW-PBE), respectively. Its deviation from various experimental measurements [28–31] is within 0.45%~1%. The calculated bulk modulus agrees with other DFT results [25,28] within small errors of 5.8% (PAW-LDA) and 2.2% (PAW-PBE). Its discrepancy with a recent experimental measurement [31] is only −1.39%. Hence, the PAW-PW91 was adopted for all the subsequent calculations.

**Table 1.** Predicted lattice constant, internal free parameter, and bulk modulus of $Y_2Zr_2O_7$ in comparison with available theoretical and experimental results.

| Method | lattice Constant $a$ (Å) | Oxygen Displacement $\delta$ | Bulk Modulus $B$ (GPa) |
|---|---|---|---|
| PAW-LDA | 10.41 | 0.330 | 209.3 |
| PAW-PW91 | 10.48 | 0.342 | 187.0 |
| PAW-PBE | 10.48 | 0.341 | 181.0 |
| PAW-LDA [25] | 10.46 | 0.342 | 176.2 |
| PAW-PW91 [26] | 10.50 | 0.342 | - |
| PAW-PBE [27] | 10.44 | - | 182.8 |
| Expt. | 10.43 [28] | - | - |
| | 10.42 [29] | - | - |
| | 10.38 [30] | - | - |
| | 10.42 [31] | - | 189.6 |

### 3.2. Atomic Modeling of Surfaces

Low miller-indexed $Y_2Zr_2O_7$ polar surfaces (100) and (110) were modeled as a semi-infinite stacking of repeatable building blocks (a set of atomic layers) as ideally cleaved from their bulk phases upon full relaxation. Figure 1 plots the as-constructed atomic structures of $Y_2Zr_2O_7$ (110) and (100) surfaces. Both of the surfaces have the smallest repeatable unit building block of [8Y-8Zr-28O], as indicated in solid red boxes. Inside these, the (4O/2Y2Zr/4O) and (4O/2Y2Zr/4O)* sub-units of the (110) are anti-symmetric w.r.t the inversion center (IC), and so are the (1O/5O/1O) and the (1O/5O/1O)* sub-units of the (100). We should note that based on the smallest repeatable building block of [8Y-8Zr-28O], the stoichiometric (110) and (100) surfaces can be constructed in two different ways—-namely, Model I and Model II, as highlighted in Figure 1. Both models differ in surface terminations, but any other construction manners are essentially equivalent to one of the two models.

The as-cleaved four models of stoichiometric (110) and (100) surfaces can all retain their anti-symmetry, but they will always possess a net non-zero perpendicular electric dipole across the vacuum thickness [32]. Such a polar-free surface cannot possibly be stable on its own. Near-surface charge redistributions arising from surface relaxations (or reconstructions) may partially reduce the surface dipoles, but, nevertheless, the surface still remains as metastable, and it can be highly sensitive to surface vacancies [33,34]. By following the approach of Sayle [35], we chose to manipulate the topmost atomic-layer structures of the as-cleaved stoichiometric (110) and (100) surfaces by anti-symmetrically removing half atoms from both of the opposite sides of the slab. In this way, a defective but neutral stoichiometric surface could be obtained with net-zero perpendicular dipole moments—-that is to say, the non-zero perpendicular electric dipoles on both of the opposite sides are exactly canceled out. We should note that even by this treatment, the (100) Model II surface cannot be made stable and is thus unlikely to be feasible, and is therefore ignored

in our calculations. Eventually, a total of four and two feasible stoichiometric terminations can be suggested for the (110) and (100) surface, respectively, and are named 110-stoi-1Y1Zr_1, 110-stoi-1Y1Zr_2, 110-stoi-1Y1Zr3O_1, and 110-stoi-1Y1Zr3O_2 for the (110) in Figure 2, and are named 100-stoi-1Y1Zr_1 and 100-stoi-1Y1Zr_2 for the (100) in Figure 3. Here, -1Y1Zr or -1Y1Zr3O records the corresponding atomic numbers and the species on the topmost terminated layer. The same nomenclature is applied to all the similar graphs hereafter.

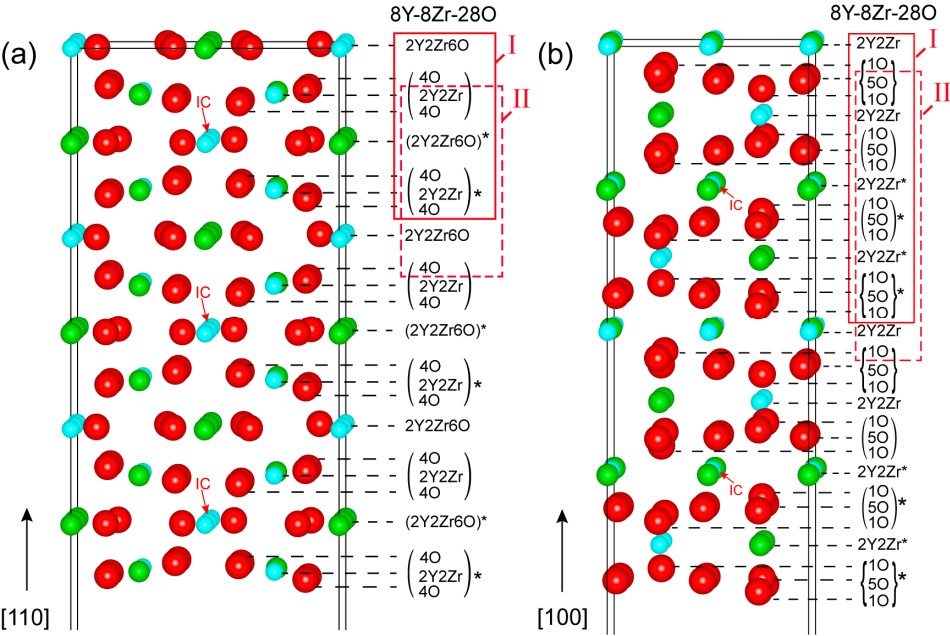

**Figure 1.** The atomic structures of $Y_2Zr_2O_7$ surfaces: (**a**) the (110) and (**b**) the (100). Here and thereafter, Y, O and Zr atoms are shown as green, red and blue balls, respectively. Note that the smaller structures with the symbol "*" in the structures are anti-symmetric to each other with respect to the inversion center (IC).

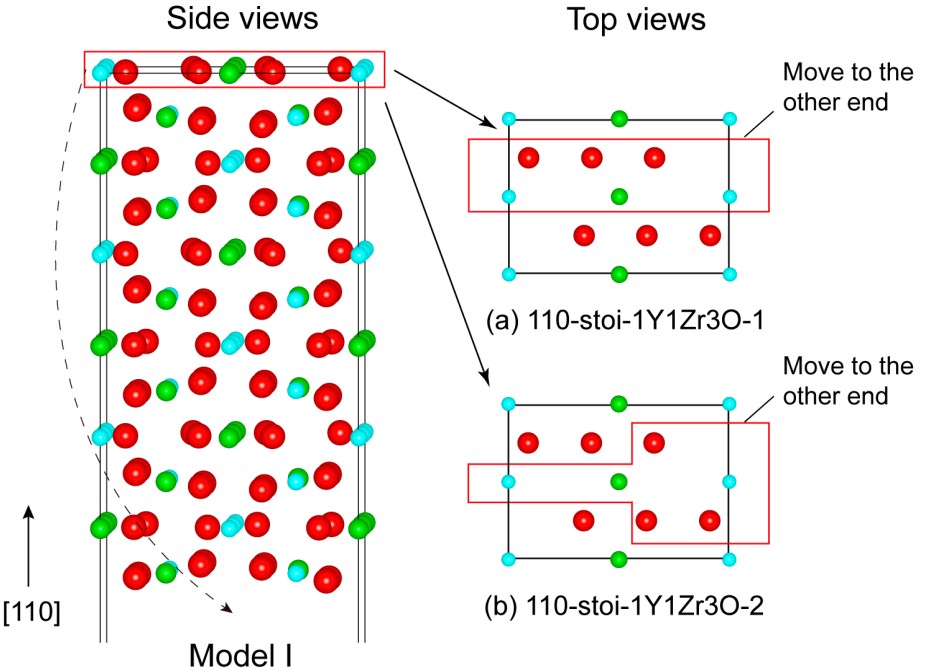

**Figure 2.** *Cont.*

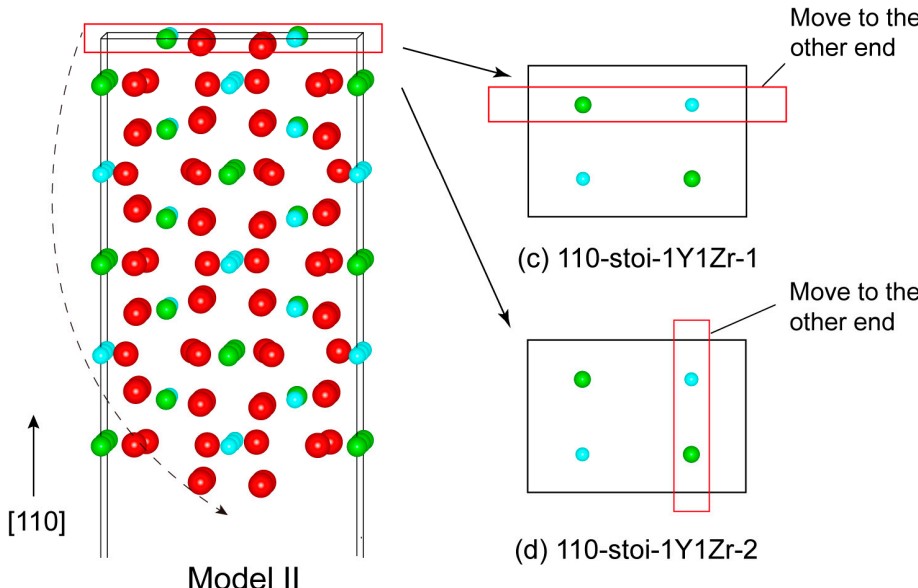

**Figure 2.** The two possibly stable structures (Model I and Model II) of the (110) surface. Depending on the manner of removing half of the surface atoms in the red box from the outermost planes, model I can be constructed with the two termination types of (**a**) 110-stoi-1Y1Zr3O_and (**b**) 110-stoi-1Y1Zr3O_2. Two other terminations for Model II are named (**c**) 110-stoi-1Y1Zr_11 and (**d**) 110-stoi-1Y1Zr_2.

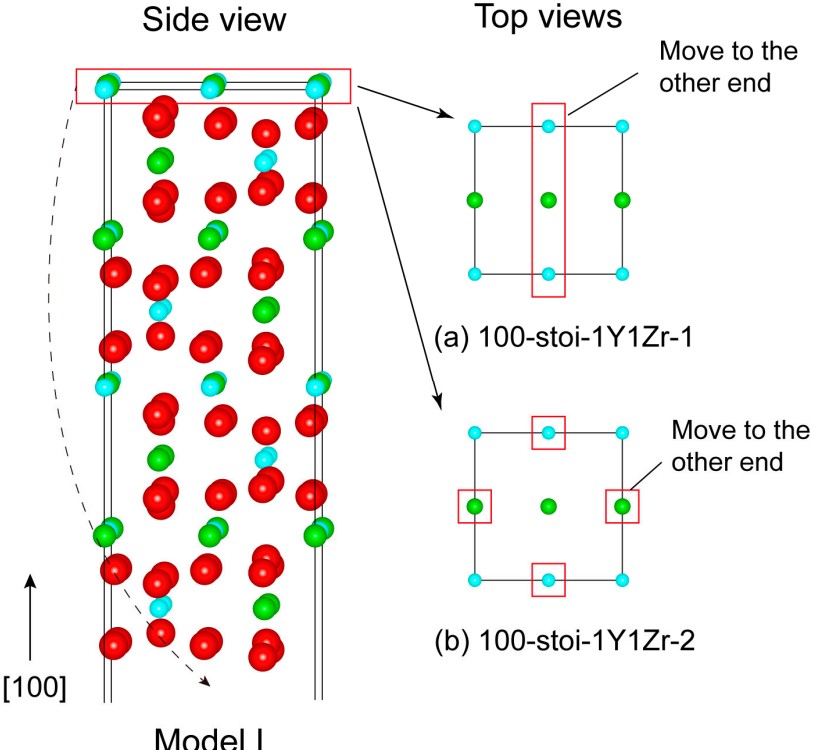

**Figure 3.** The only possibly stable structure (Model I) of the (100) surface with the two termination types of (**a**) 100-stoi-1Y1Zr_1 and (**b**) 100-stoi-1Y1Zr_2. Note the manner is the same as the mentioned above.

Similarly, we can further construct various non-stoichiometric surfaces of the (110) and (100). Figure 4 plots the four feasible non-stoichiometric structures of the (110) surface, i.e., 110-ns-2Y2Zr, 110-ns-4O_1, 110-ns-2Y2Zr6O, and 110-ns-4O_2. Figure 5 plots the three feasible non-stoichiometric structures of the (100) surface, i.e., 100-ns-2Y2Zr, 100-ns-1O,

and 100-ns-5O. For all of these non-stoichiometric structures, the anti-symmetry can be well retained. Non-zero perpendicular dipole moments are equal and opposite on both of the two opposite sides of the surface slabs, which can be exactly cancelled out—-that is to say, these non-stoichiometric surface structures can all feasibly be stable, with no need for surface vacancies.

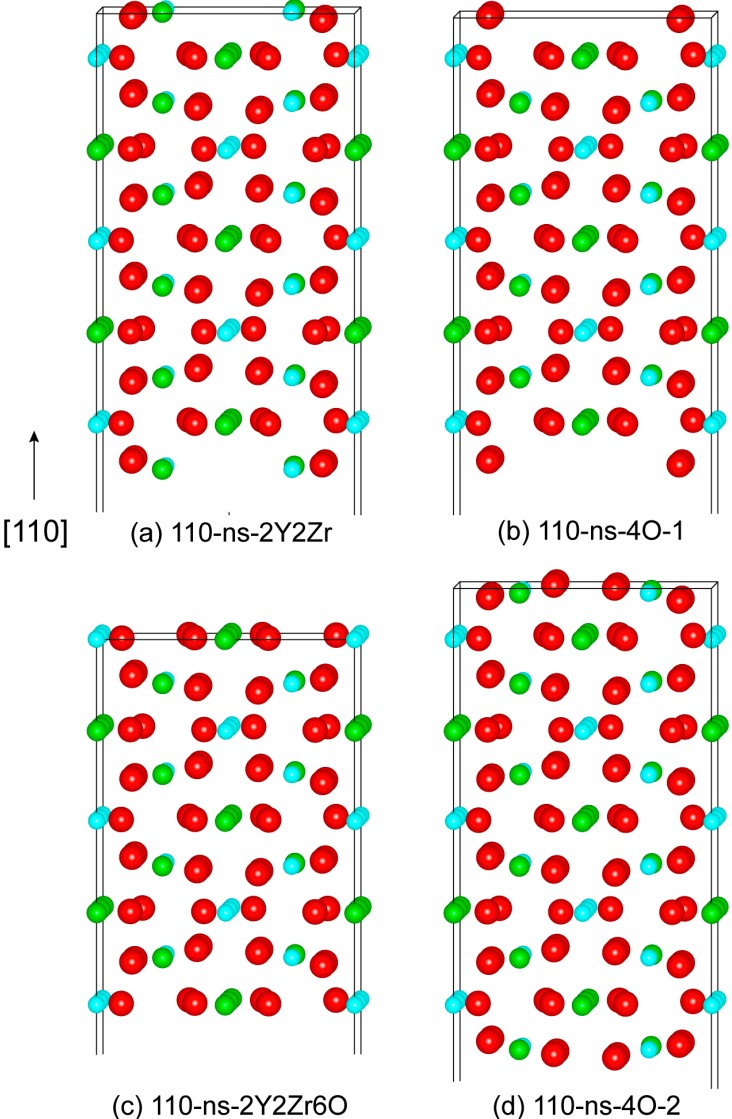

**Figure 4.** The four possibly stable non-stoichiometric structures of the (100) surface.

### 3.3. Surface Formation Energy and Relative Stability

Based on a thermodynamic defect model, we calculated the formation energies of surface structures with all possible terminations. Based on the calculated surface formation energies, their relative structural stabilities were further evaluated. The surface formation energy, $\gamma$, was calculated as [36]:

$$\gamma = \frac{1}{2A}\left(E_{tot,surf} - E_{tot,bulk} + P\Delta V - T\Delta S\right) \tag{1}$$

where $E_{tot,surf}$ is the calculated total energy of the fully relaxed surface supercell; $A$ is the cross-sectional area of the supercell; $E_{tot,bulk}$ is the calculated total energy of the bulk-phase supercell consisting of the same number of atoms; $\Delta V$ is the volume change due to surface relaxation w.r.t its bulk structure, which is generally ignorable under the environment

pressure $P$; and $\Delta S$ is mostly the vibrational entropy change induced by the surface cleavage from the bulk structure, which is generally small. The term of $-T\Delta S$ can lower surface formation energy as $T$ increases, yet its contribution to surface energy and their relative ordering is negligible at $T$ far below the metaling point. As a general convention, the last two terms of $P\Delta V$ and $T\Delta S$ in Equation (1) are often disregarded.

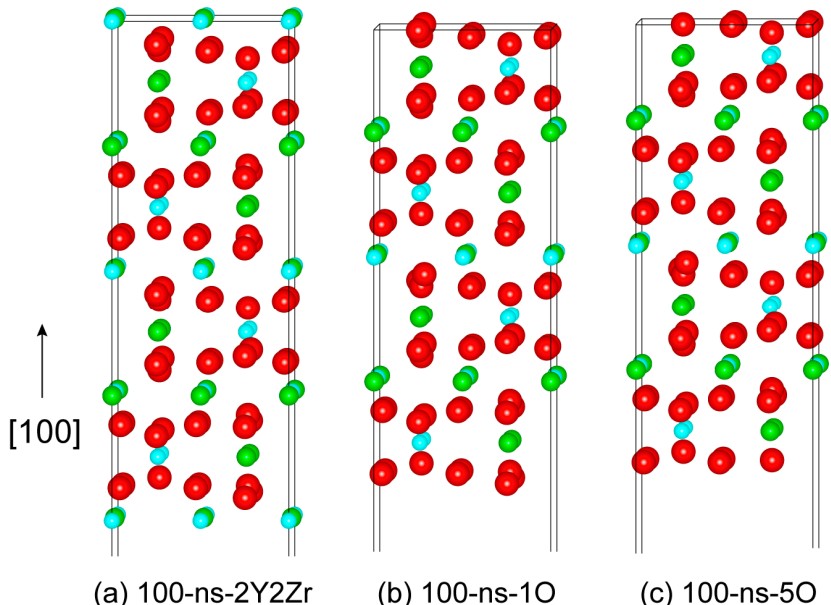

**Figure 5.** The three possibly stable non-stoichiometric structures of the (100) surface.

Table 2 compares the unrelaxed and relaxed surface formation energies of stoichiometric (110) and (100). The positive values of all of these surface energies reflect the general fact that energy input is always required for creating a surface. A lower surface energy suggests a relatively higher stability. In this sense, the relative stabilities of the unrelaxed stoichiometric (110) surfaces follows the ordering of 1Y1Zr_1 > 1Y1Zr3O_1 > 1Y1Zr3O_2 > 1Y1Zr_2, and surface relaxation would change the ordering to 1Y1Zr3O_2 > 1Y1Zr_2 > 1Y1Zr3O_1 > 1Y1Zr_1. For the stoichiometric (100) surfaces, though, 100-stoi-1Y1Zr_1 is predicted as more energy-favored after relaxation. No direct comparison with experiments or with other calculations is yet available, but it is nevertheless clear that surface relaxation has a critical role in stabilizing surface structures.

**Table 2.** Calculated formation energies of stoichiometric (100) and (110) surfaces.

| Termination Type | Surface Formation Energy (J/m$^2$) | |
| :---: | :---: | :---: |
| | **Unrelaxed** | **Relaxed** |
| 110-stoi-1Y1Zr_1 | 3.36 | 1.59 |
| 110-stoi-1Y1Zr_2 | 3.97 | 1.33 |
| 110-stoi-1Y1Zr3O_1 | 3.62 | 1.49 |
| 110-stoi-1Y1Zr3O_2 | 3.65 | 1.20 |
| 100-stoi-1Y1Zr_1 | 4.20 | 1.47 |
| 100-stoi-1Y1Zr_2 | 3.44 | 1.75 |

Under certain oxygen partial pressures and temperatures, non-stoichiometric surfaces, featured with off-stoichiometric compositions of $N_Y:N_{Zr}:N_O \neq 2:2:7$, can form in equilibrium with a relatively insufficient supply of O, Zr, or Y. These surfaces could have even lower formation energies than the 110-stoi-1Y1Zr surface. As mentioned above, we have constructed a total of four non-stoichiometric (110) surfaces with ideal terminations of 110-ns-2Y2Zr, 110-ns-2Y2Zr6O, 110-ns-4O_1, and 110-ns-4O_2 in Figure 4, and, similarly,

we have constructed a total of three non-stoichiometric (100) surfaces with ideal terminations of 100-ns-2Y2Zr, 100-ns-1O, and 100-ns-5O in Figure 5. In order to investigate the formation preferences and stabilities of these non-stoichiometric surfaces, we restricted the calculations below to ideal terminations, and we also ignored any possible intermediate transition state between two ideal terminations that may tentatively induce certain numbers of surface vacancies.

The formation and the relative stabilities of all these non-stoichiometric (110) and (100) surfaces would depend sensitively on the thermodynamic conditions of ambient environment. A more general formula for the surface formation energy can be deduced as [37]:

$$\gamma = \frac{1}{2A}\left(E_{tot,surf} - N_Y\mu_{Y,surf} - N_{Zr}\mu_{Zr,surf} - N_{O_2}\mu_{O_2,gas} + P\Delta V - T\Delta S\right) \qquad (2)$$

where $N_i$ is the atom number of element, $i$ in the surface supercell, and $\mu_i$ is the elemental chemical potential. Equation (2) is generally valid for both the stoichiometric and the non-stoichiometric surfaces. Once again, the last two terms of $P\Delta V$ and $T\Delta S$ in Equation (2) can be disregarded as the first order of approximation. Furthermore, under the thermodynamic equilibrium, the chemical potential of crystalline $Y_2Zr_2O_7$ shall equal the sum of all of the elemental potentials, $\mu_i$, i.e., $\Delta G = 0$, for either oxide decomposition or formation:

$$2\mu_{Y,surf} + 2\mu_{Zr,surf} + \frac{7}{2}\mu_{O_2,gas} = \mu_{Y_2Zr_2O_7,surf} = \mu_{Y_2Zr_2O_7,bulk} \qquad (3)$$

Furthermore, it can be noticed that all of the surface models in Figures 4 and 5 have an equi-atomic ratio of $Y$ and $Zr$, i.e., $N_Y = N_{Zr}$. Therefore, for these surface structures, Equation (2) can be rewritten as:

$$\begin{aligned}\gamma &= \frac{1}{2A}\left[E_{tot,surf} - N_Y\left(\mu_{Y,surf} + \mu_{Zr,surf}\right) - N_{O_2}\mu_{O_2,gas}\right] \\ &= \frac{1}{2A}\left[E_{tot,surf} - N_Y\left(\frac{1}{2}\mu_{Y_2Zr_2O_7,bulk} - \frac{7}{4}\mu_{O_2,gas}\right) - N_{O_2}\mu_{O_2,gas}\right]\end{aligned} \qquad (4)$$

Here, $\mu_{O_2,gas}$ can be further related with the chemical potentials of elements $Y$ and $Zr$ and the crystalline $Y_2Zr_2O_7$, through the formation enthalpy of $Y_2Zr_2O_7$, $\Delta H_{f,Y_2Zr_2O_7}$ according to the reaction of $2Y + 2Zr + \frac{7}{2}O_2 \rightarrow Y_2Zr_2O_7$. Hence:

$$\begin{aligned}\mu_{O_2,gas} &= \mu^0_{O_2,gas} + kT\ln\frac{p_{O_2}}{p^0_{O_2}} \\ &= \left(\frac{2}{7}\mu^0_{Y_2Zr_2O_7,bulk} - \frac{4}{7}\mu^0_{Y,bulk} - \frac{4}{7}\mu^0_{Zr,bulk} - \frac{2}{7}\Delta H^0_{f,Y_2Zr_2O_7}\right) + kT\ln p_{O_2}\end{aligned} \qquad (5)$$

where the superscript '0' represents the standard state of each quantity. With Equations (4) and (5), Equation (2) can be further written as a function of the environmental conditions of $P_{O2}$ and $T$:

$$\begin{aligned}\gamma = \quad &\frac{1}{2A}\big[E_{tot,surf} - \frac{1}{2}N_Y\mu_{Y_2Zr_2O_7,bulk} + (\frac{1}{2}N_Y - \frac{2}{7}N_{O_2})(\mu^0_{Y_2Zr_2O_7,bulk} \\ &-2\mu^0_{Y,bulk} - 2\mu^0_{Zr,bulk} - \Delta H^0_{f,Y_2Zr_2O_7}) + (\frac{7}{4}N_Y - N_{O_2})kT\ln p_{O_2}\big]\end{aligned} \qquad (6)$$

Due to the fact that the large cancellation occurs between $\Delta H^0_{f,Y_2Zr_2O_7}$ and all the chemical potentials $\mu^0_i$ and also between the total energy term $E_{tot,surf}$ and $N_Y\mu_{Y_2Zr_2O_7}$, their temperature dependences can be assumed to largely cancel out. For this reason, as the first order of approximation, the non-stoichiometric surface formation energy can be approximated by zero-kelvin energetics calculations, so that the temperature effect would only arise from the surface non-stoichiometry due to $N_Y/N_O \neq 2/7$. We should note that

$\Delta H_{f,Y_2Zr_2O_7}^0$ in Equation (6) is still unknown in experiments, but that it can be approached using Hess's law by considering the following three step reactions:

$$
\begin{aligned}
4Y + 3O_2 &\rightarrow 2Y_2O_3 & 2\Delta H_1(0\text{ K}) \\
4Zr + 4O_2 &\rightarrow 4ZrO_2 & 4\Delta H_2(0\text{ K}) \\
+) \quad 2Y_2O_3 + 4ZrO_2 &\rightarrow 2Y_2Zr_2O_7 & 2\Delta H_3(0\text{ K}) \\
\hline
4Y + 4Zr + 7O_2 &\rightarrow 2Y_2Zr_2O_7 & 2\Delta H_f(0\text{ K})
\end{aligned}
$$

Therefore:

$$\Delta H_{f,Y_2Zr_2O_7}^0 \approx \Delta H_f(0\text{ K}) = \Delta H_1(0\text{ K}) + 2\Delta H_2(0\text{ K}) + \Delta H_3(0\text{ K}) \tag{7}$$

Here, we calculated $\Delta H_3(0\text{ K})$ as $\sim -4.01$ eV ($\sim -386.6$ kJ/mol). Knowing the experimental data of $\Delta H_1$ ($-1905.3$ kJ/mol) and $\Delta H_2$ ($-1100.6$ kJ/mol) at 298 K as well as the specific heat capacities of $Cp(Y_2O_3) = 102.5$ and $Cp(ZrO_2) = 56.2$ J/(deg·mol) [38], we deduced $\Delta H_1(0\text{ K})$ and $\Delta H_2(0\text{ K})$ as approximately $-20.08$ eV ($\sim -1935.7$ kJ/mol) and $-11.59$ eV ($\sim -1117.3$ kJ/mol), respectively. Finally, we obtained $\Delta H_{f,Y_2Zr_2O_7}^0 = -47.27$ eV ($\sim -4556.9$ kJ/mol).

Combining Equations (6) and (7), we calculated the formation energies of all of the non-stoichiometric (110) and (100) surfaces within a wide range of $P_{O_2}$ for $T = 300$ and 1400 K in comparison with all of the stoichiometric counterparts in Figures 6 and 7. Again, it should be noted that the high temperature of 1400 K was chosen for the fact that it is close to the practical sintering temperature of $Y_2Zr_2O_7$ and, also, to the conventional annealing temperature of NFAs.

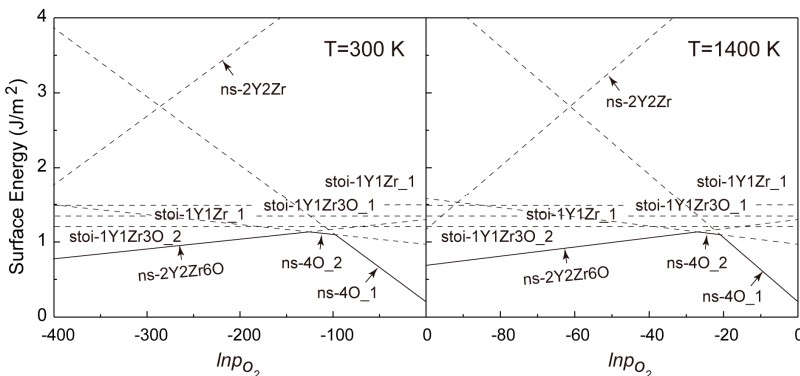

**Figure 6.** Calculated formation energies of various termination types of the $Y_2Zr_2O_7$ (110) w.r.t $p_{O2}$ for $T = 300$ K and 1400 K.

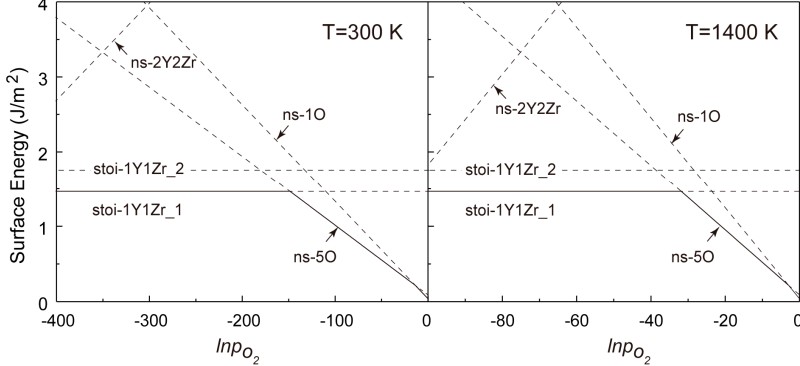

**Figure 7.** Calculated formation energies of various termination types of the $Y_2Zr_2O_7$ (100) w.r.t. $p_{O2}$ for $T = 300$ K and 1400 K.

It is revealed in Figure 6 that for the (110) surfaces, the formation energies of all non-stoichiometric terminations can be lower than those of the stoichiometric ones in a certain range of $p_{O2}$ at $T$ = 300 and 1400 K. This clearly deviates from the general belief of most metal oxides that a non-stoichiometric surface will always be less stable owing to the presence of more unsaturated chemical bonds at its topmost surface layers. As seen in Figure 6, increasing $p_{O2}$ certainly favors changing the most stable termination from the (Y,Zr)-rich (110-ns-2Y2Zr6O) to the O-rich ones (110-ns-4O_2 and 110-ns-4O_1), and no stoichiometric termination can be expected for the (110). The critical $p_{O2}$ for the termination transitions move to higher values at a higher $T$, and the 110-ns-2Y2Zr6O termination always dominates over a wider range of $p_{O2}$. In contrast, for the (100) surfaces in Figure 7, the most stable termination changes from the stoi-1Y1Zr_1 to the O-rich ns-5O as increasing $p_{O2}$. Again, the critical $p_{O2}$ for termination transition moves to higher values as increasing $T$, and the stoi$-$1Y1Zr_1 termination dominates over a wider range of $p_{O2}$. The surface phase diagrams in Figures 6 and 7 can be used to predict the most energy-favored (110) and (100) surface structures during the growth of $Y_2Zr_2O_7$ under a given thermodynamic condition of $T$ and $p_{O2}$. The surface structures and the thermodynamics can also provide an important knowledge base for future studies on $Y_2Zr_2O_7$ interfaces in NFAs.

## 4. Conclusions

The atomic structure and the relative stabilities of $Y_2Zr_2O_7$ (100) and (110) surfaces have been extensively investigated using DFT energetics calculations. Using a thermodynamic defect model, the formation energies of non-stoichiometric surfaces were expressed as a function of oxygen partial pressure ($p_{O2}$) and temperature ($T$). All of the possible stoichiometric and non-stoichiometric surface structures were considered with various ideal terminations. Our calculations suggested that surface formation energy and its relative stability sensitively depend on surface stoichiometry, which, in turn, is thermodynamically determined by the environment conditions of $T$ and $p_{O2}$. With increasing $p_{O2}$, the most stable termination of the $Y_2Zr_2O_7$ (110) surfaces changes from the ns-2Y2Zr6O to the O-rich ns-4O_2 and ns-4O_1, while that of the $Y_2Zr_2O_7$ (100) surface changes from the stoi-1Y1Zr_1 to the O-rich ns-5O. The critical $p_{O2}$ for the termination transitions moves to higher values as increasing $T$.

**Author Contributions:** Conceptualization, H.W. and Y.J.; methodology, H.W.; software, H.W.; validation, H.W., Y.J. and Y.W.; formal analysis, H.W.; investigation, H.W.; resources, H.W.; data curation, H.W. and J.Z.; writing—original draft preparation, H.W.; writing—review and editing, H.W. and Y.J.; visualization, H.W.; supervision, J.Z. and J.S.; project administration, Y.J.; funding acquisition, Y.J. All authors have read and agreed to the published version of the manuscript.

**Funding:** This research was funded by [the National MCF Energy R&D Program of China] grant number [2018YFE0306100] and [the National Science Foundation of China] grant number [No. 52001331].

**Institutional Review Board Statement:** Not applicable.

**Data Availability Statement:** Not available.

**Acknowledgments:** The authors would like to thank the financial support from the Science and Technology on Reactor Fuel and Materials Laboratory of China. We would also like to acknowledge the computational resources at the High Performance Computing Center of Central South University and Hefei Advanced Computing Center are highly appreciated.

**Conflicts of Interest:** The authors declare no conflict of interest.

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
