# Peer review of "Stoichiometric and Nonstoichiometric Surface Structures of Pyrochlore Y2Zr2O7 and Their Relative Stabilities: A First-Principles Investigation"

_coatings, doi:10.3390/coatings13071203_

Round 1

Reviewer 1 Report

Present manuscript by Wu et al., deals with the investigations on the surface stability of Y2Zr2O7 pyrochlore phase. This theoretical investigation is important and will be helpful to the future experimental investigation if any. Article is well written and can be accepted for publication after considering few minor comments below. 

(1) What is critical termination point? It has to be mentioned in text somewhere. 

(2) What is the importance of the present investigation ? It should be highlight more in the introduction. 

(3) Why pO2 influences similarly to the critical termination behaviour of different surfaces?  

(4) What is the mechanism of different crystallographic surfaces react differently to the pO2? 

(5) Authors must crosscheck the correctness of using 'ns-2Y2Zr6O' in the abstract. 

Author Response

Dear Editors,

We highly appreciate the careful reading of our manuscript and the constructive comments of the editors and reviewers. We are now submitting the revised manuscript for publication reconsideration at Coatings. Enclosed please find our response to the reviewers’ comments. All relevant changes have been highlighted in red in the revised manuscript.

Very Sincerely,

Yong Jiang (THE CORRESPONDING AUTHOR)

Dr. Prof.

Materials Surface and Interface Group

Central South University

Changsha, China

EMAIL: yjiang@csu.edu.cn

Reviewers' comments:

Reviewer #1

Present manuscript by Wu et al., deals with the investigations on the surface stability of Y2Zr2O7 pyrochlore phase. This theoretical investigation is important and will be helpful to the future experimental investigation if any. Article is well written and can be accepted for publication after considering few minor comments below.

Authors: We highly appreciate the careful reading of our manuscript and the positive comments of the reviewer.

  • What is critical termination point? It has to be mentioned in text somewhere.

Authors: Thanks for the reminder. In the manuscript, we used “the transition point” to mention the critical pO2 value that corresponds to the termination transition under a given specific T. For example, for the (100) surfaces in Fig. 7, with increasing pO2, the most stable surface termination would change from the (Y,Zr)-rich (stoi-1Y1Zr_1) to the O-rich one (ns-5O) at the critical value of pO2=e-150 at T=300K. While at T=1400K, the same termination transition would occur at a higher critical value of pO2=e-31. To avoid any possible misleading, we change to the new term - the critical pO2 value for termination transition – in the revised manuscript (Page 10), for predicting the surface termination transition at different temperatures.

  • What is the importance of the present investigation? It should be highlight more in the introduction.

Authors: Thanks for this suggestion. Recent experiments and DFT calculations have suggested that Y-Zr-O would be the most favored and abundant oxide nano-phase in NFAs with Zr addition [R1, R2]. Our results in the present work can be used to not only predict the most energy-favored equilibrium (110) and (100) surface structures during the growth of Y2Zr2O7 under a given thermodynamic condition, which provides a critical knowledge base for our future investigation on Y2Zr2O7 interfaces in NFAs, but also improve our understanding of the reactivity and performance of Y2Zr2O7 surfaces in various high temperature catalytic applications including SOFC. This significance has been emphasized at the end of the introduction.

  • Why pO2 influences similarly to the critical termination behavior of different surfaces?

Authors: Thanks of raising this issue. The equilibrium termination structure of a metal-oxide surface can be either metal-rich, stoichiometric, or O-rich. According to Eq. (6), a metal-rich termination (NY/NO > 2/7) would have an increasing surface formation energy, and an O-rich termination (NY/NO < 2/7) would have a decreasing surface formation energy with the increasing pO2 at any given T. These trends reflect the fundamental dependence of a metal-oxide surface termination on thermodynamic conditions (pO2 and T) which must be obeyed by all kinds of metal-oxides, and have been clearly shown by the results in Figs. 6 and 7. A nature consequence of these trends is that a metal-rich termination would dominate in lower pO2 values and an O-rich termination would dominate in higher pO2 values at any given T. Hence, tuning T and/or pO2 can provide us a freedom for manipulating the formation preference and relative stabilities of difference termination structures, but the fundamental dependence of termination structures on thermodynamic conditions (pO2 and T) must be first clarified. This present work is aimed to predict the critical pO2 values that correspond to surface termination transitions from metal-rich to stoichiometric or O-rich at different T, and the surface phase diagrams (Figs. 6 and 7) can be thus constructed accordingly, to quantitatively reveal the dependence of surface termination on pO2 and T. That is, different Y2Zr2O7 surfaces always tend to move from a metal-rich (or O-deficient) termination to an O-rich termination as increasing pO2 at a given T, and our efforts in this work are devoted to determine the critical pO2 values for termination transitions.

  • What is the mechanism of crystallographic surfaces react differently to the pO2?

Authors: Please kindly refer to our answer to question 3.

  • Authors must crosscheck the correctness of using ‘ns-2Y2Zr6O’ in the abstract.

Authors: This is correct. The ns-2Y2Zr6O here is to mention the (Y,Zr)-rich termination structure of 110-ns-2Y2Zr6O as schematically shown in Fig. 4c.

Again, we highly appreciate all the reviewers for their careful reading of our manuscript. All these comments and suggestion are very thoughtful and helpful, and has greatly improved the quality and scope of this manuscript.

Reviewer 2 Report

Jiang and his group have studied the surfaces of Y2Zr2O7 from DFT calculations and the manuscript is very interesting, use correct methods and is well done. It can be accepted for publication with only one minor correction.

The authors use several functionals in the study, GGA, PW91, PBE, etc. There are not justification for this collection, they should include a paragraph to describe the utility of all methods, the advantages or problems found in each case and they should assure which are the best results and why they consider them the best option.

-       The manuscript deals with the oxide combined system of yttrium and zirconium, it would be useful to clarify in the introduction which other systems with other elements exist and why the yttrium-zirconium is the most important for this group.

-       The authors calculate the enthalpy of formation of the combined oxide by using the Hess law and obtain an excellent negative result, but they do not give the corresponding values of the yttrium and zirconium oxides, only make the reference to the published values (ref. 34). They should show in the present work these values in order to appreciate the thermodynamic effect of the formation of the combined oxide.

I think this is an excellent work and the authors present enough information and results to give a good comment, for this reason I make the recommendation to accept the manuscript with minor revision. I hope this information will be useful for you.

Author Response

Dear Editors,

We highly appreciate the careful reading of our manuscript and the constructive comments of the editors and reviewers. We are now submitting the revised manuscript for publication reconsideration at Coatings. Enclosed please find our response to the reviewers’ comments. All relevant changes have been highlighted in red in the revised manuscript.

Very Sincerely,

Yong Jiang (THE CORRESPONDING AUTHOR)

Dr. Prof.

Materials Surface and Interface Group

Central South University

Changsha, China

EMAIL: yjiang@csu.edu.cn

Reviewers' comments:

Reviewer #2

Jiang and his group have studied the surfaces of Y2Zr2O7 from DFT calculations and the manuscript is very interesting, use correct methods and is well done. It can be accepted for publication with only one minor correction.

Authors: We highly appreciate the careful reading of our manuscript and the positive comments of the reviewer.

- The authors use several functionals in the study, GGA, PW91, PBE, etc. There are not justification for this collection, they should include a paragraph to describe the utility of all methods, the advantages or problems found in each case and they should assure which are the best results and why they consider them the best option.

Authors: The optimal choice of XC-functionals was determined by best reproducing the experimental bulk structures and elastic modulus in literatures. All the results are compared in Table 1. From which, it is clear that all our calculation results agree favorably with available theoretical and experimental values. Especially, the PAW-PW91 yielded the best predictions on bulk lattice constant, bulk modulus, and the internal free parameter. The PAW-PW91 calculated lattice constant deviates slightly from previous DFT predictions [26–28], by only 0.2% (PAW-LDA), -0.2% (PAW-PW91), and 0.4% (PAW-PBE), respectively. Its deviation from various experimental measurements [29–32] is within 0.45~1%. The PAW-PW91 calculated bulk modulus agrees with the previous calculations [20, 23] within errors of 5.8% (PAW-LDA) and 2.2% (PAW-PBE). Its discrepancy with a recent experimental measurement [32] is -1.39 % only. Thus, the PAW-PW91 was adopted to ensure the accuracy of all the subsequent calculations.

-  The manuscript deals with the oxide combined system of yttrium and zirconium, it would be useful to clarify in the introduction which other systems with other elements exist and why the yttrium-zirconium is the most important for this group.

Authors: Thanks for bringing up this issue. Y-Ti-O is the most popular nano-oxide phase in NFAs and has attracted most research efforts in the latest decade. In recent years, various different complex Y-X-O nano-oxides have been introduced into NFAs, and among which, Y-Zr-O nano-oxides show great promises. Several experiments have suggested Zr alloying can induce finer nano-oxides, for instance, Ti+Zr co-alloying refined Y-Ti-O nano-oxides to less than 10 nm on average [R3-R5]. Y+Zr co-alloying in FeCrAl-based NFAs can greatly reduce the number of coarse Y-Al-O nano-oxides in favor for finer Y-Zr-O nano-oxides [R6-R9]. One of our previous first-principles studies on nano-clusters (NCs) nucleation also suggested that Y-Zr-O NCs nucleate in preference to Y-Ti-O and Y-Al-O NCs in ferritic matrix [R10]. Hence in this work, we proposed to first investigate the surface structures and energetics of pyrochlore Y2Zr2O7, which is a prerequisite for the fundamental understanding of their interface structures and properties in NFAs. We have revised the introduction session accordingly, to clarify our motivation in this work.

-  The authors calculate the enthalpy of formation of the combined oxide by using the Hess law and obtain an excellent negative result, but they do not give the corresponding values of the yttrium and zirconium oxides, only make the reference to the published values (ref. 34). They should show in the present work these values in order to appreciate the thermodynamic effect of the formation of the combined oxide.

Authors: Thanks for pointing out this. We have added these values immediately after Eq. (7) in Page 9.

I think this is an excellent work and the authors present enough information and results to give a good comment, for this reason I make the recommendation to accept the manuscript with minor revision. I hope this information will be useful for you.

Authors: We thank again for the positive comments of the reviewer.

Again, we highly appreciate all the reviewers for their careful reading of our manuscript. All these comments and suggestion are very thoughtful and helpful, and has greatly improved the quality and scope of this manuscript.

Reviewer 3 Report

The article is describing the surface structure of pyrochlore Y2Zr2O7. This is interesting material which can be in future applied for various high temperature catalytic application including SOFC.

The paper can be recommended for publication

Author Response

Dear Editors,

We highly appreciate the careful reading of our manuscript and the constructive comments of the editors and reviewers. We are now submitting the revised manuscript for publication reconsideration at Coatings. Enclosed please find our response to the reviewers’ comments. All relevant changes have been highlighted in red in the revised manuscript.

Very Sincerely,

Yong Jiang (THE CORRESPONDING AUTHOR)

Dr. Prof.

Materials Surface and Interface Group

Central South University

Changsha, China

EMAIL: yjiang@csu.edu.cn

Reviewers' comments:

Reviewer #3

The article is describing the surface structure of pyrochlore Y2Zr2O7. This is interesting material which can be in future applied for various high temperature catalytic application including SOFC. The paper can be recommended for publication.

Authors: We highly appreciate the positive comments of the reviewer. The information about many other possible application of Y2Zr2O7 is very useful and we have emphasized this significance into the introduction session.
